# Feasibility Analysis of Polyurethane-Prepolymer-Modified Bitumen Used for Fully Reclaimed Asphalt Pavement (FRAP)

**DOI:** 10.3390/ma16165686

**Published:** 2023-08-18

**Authors:** Minggang Sun, Jianling Wang, Hongpeng Sun, Bin Hong

**Affiliations:** 1Long Jian Road & Bridge Co., Ltd., 109 Songshan Road, Nangang District, Harbin 150090, China; 2School of Transportation Science and Engineering, Harbin Institute of Technology, 73 Huanghe Road, Nangang District, Harbin 150090, China

**Keywords:** polyurethane-prepolymer-modified bitumen, fully reclaimed asphalt pavement, design, rheological property, tensile property, economic analysis

## Abstract

Asphalt pavement recycling technology with high reclaimed asphalt pavement (RAP) content has always been limited by unsatisfactory pavement performance and the rising cost of pavement materials. To address these challenges, polyurethane-prepolymer-modified bitumen (PPB) was proposed to be utilized as the asphalt binder of fully reclaimed asphalt pavement (FRAP) in this study. The proper formula of the PPB binder was determined based on a range of tests. The rheological behavior and tensile properties of the PPB binder were then investigated, and the economic cost of materials was discussed as well. Results revealed that the PPB system can be obtained through chemical synthesis using readily available raw materials. The reaction of polyurethane prepolymer and chain extender provides PPB with significant improvement in temperature susceptibility, rutting resistance, and tensile properties. It is also demonstrated in this study that the PPB mixture containing 100% RAP, on the whole, takes advantage of cost-saving especially compared to the epoxy asphalt mixture. Therefore, the PPB binder exhibits a favorable application prospect in FRAP.

## 1. Introduction

In the past few decades, RAP has been gradually implemented during the maintenance process for seriously damaged asphalt pavement [1,2,3] due to its potential to save costs, reduce greenhouse gas emissions, and promote the recycling of natural resources [4,5,6]. However, the poor quality of RAP mixtures is an important limiting factor for maximizing RAP content in asphalt pavement. For instance, the RAP content for hot recycled asphalt pavement is usually no more than 35% [7,8], while for a cold recycled emulsified asphalt mixture, the RAP content is relatively higher [9]. Moreover, the addition of RAP has a great influence on the performance and service life of recycled pavement, as high RAP content may cause adverse effects on the water stability, cracking resistance, and fatigue resistance of the recycled mixture [10,11]. Therefore, the finite RAP content and insufficient service life of the RAP mixture have been identified as major challenges that hinder the advancement of recycling technology in asphalt pavement.

For a long time, the quality of the RAP mixture could not be ensured due to the difficulty in integrating the aged and unaged asphalt binder. Two mainstream theories that have been widely accepted among researchers to explain the mechanism of integrating aged and unaged binders and aggregates are the following: the colloidal compatibility theory and the component conditioning theory [12,13]. The former suggests that the asphalt binder is a colloidal system consisting of asphaltenes (solutes) and soft asphaltenes (solvents). Its stability mainly depends on asphaltene content and solubility parameters differences in asphaltenes and soft asphaltenes [12]. The latter proposes that the aging of asphalt binder in RAP brings changes in the proportions of asphalt components, making it hard to meet the requirements of compatibility [13].

Out of the above-mentioned situation, the use of asphalt rejuvenators or recycling agents is a common solution in the road paving industry to enhance the cracking resistance of recycled asphalt mixtures. However, one concern associated with these rejuvenators is the potential for gradual change, which may necessitate the addition of new virgin binders and aggregates [14]. Meanwhile, the utilization of RAP in the warm mix asphalt (WMA) technology has gained attention in many countries. Compared to the hot mix asphalt (HMA) technology, the lower working temperature implies lower binder aging, thus allowing higher RAP content in recycled asphalt mixture. The WMA technology may have a detrimental influence on the rutting resistance of the mixture, which can be alleviated by increasing RAP content and adding rejuvenators [15]. Lorenzo I. et al. [16] predicted the long-term performance of an existing warm recycled highway pavement in Italy on the basis of the simplified viscoelastic continuum damage (S-VECD) testing approach. The results demonstrated that the WMA pavement possesses longer service life than the HMA pavement. The in situ Falling Weight Deflectometer (FWD) tests and laboratory tests on extracted cores conducted on Italian highways exhibit similar consequences [17]. However, the lower mixing temperatures used in the production of recycled asphalt mixtures make it difficult for the additives, especially the rejuvenator, to diffuse and interact with the aged asphalt [18]. The resistance to moisture damage and permanent deformation of asphalt mixtures may degrade equally because of the wet aggregates and the lack of electrochemical tendency between the binder and aggregate surface at low mixing temperatures [19].

Polyurethane prepolymer (PUP) resin refers to a low-molecular reactive midbody with isocyanate (–NCO) groups at the end and carbamate (–NHCOO–) in the molecular chain, mainly synthesized by polyols and isocyanates [20]. It is generally acknowledged that isocyanate groups are able to react with chemical components like carboxylic acid, amine, mercaptan, pyridine, thiophene, and other functional groups in bitumen [21]. As PUP possesses excellent mechanical properties, high resilience, flexibility, and chemical stability [22], PUP as an asphalt modifier has aroused increasing attention in the field of pavement engineering. Unlike styrene butadiene styrene (SBS) or ethylene vinyl acetate copolymer (EVA)-modified bitumen, PPB is an asphalt binder with excellent mechanical properties consisting of the virgin binder, high content PUP, chain extender, and other auxiliary agents [23,24]. In the PPB colloidal system, PUP will occur irreversible chemical reactions with the chain extender, water molecules, and asphalt components, producing a continuous crosslinking grid structure [25,26]. In addition, previous studies have verified that PUP and its mixture can prominently improve asphalt pavement performances. CO_2_-based environmentally friendly polyurethane and polypropylene carbonate diol (PPC) were adopted by Gong X et al. [27] to modify asphalt. The findings revealed that the storage stability and high-temperature properties of PPC-PU-modified asphalt increased with curing time. Yang T et al. [21] in situ synthesized polyurethane-modified asphalt by the one-shot process with varying dosages of polyurethane and isocyanate index *R* (–NCO/–OH). They discovered that a higher *R* value brings higher high-temperature performance, elastic recovery performance, and viscosity but leads to lower low-temperature performance. Xia L et al. [28] found that the incorporation of renewable castor oil could improve both high- and low-temperature properties, and castor oil–polyurethane prepolymer (C-PU) could be dispersed homogenously in the asphalt. Xu C et al. [29] synthesized reactive elastomer terpolymer (RET)-modified asphalt binder with PUP, the results of which revealed that PUP improved the low-temperature performance of asphalt. Due to these advantages, polyurethane-modified asphalt became an increasingly prevalent material applied in waterproofing, road paving, and crack maintenance.

Using PPB as a binder for fully reclaimed asphalt pavement instead of new asphalt can improve pavement performance, which may be an effective way to achieve recycled asphalt pavement containing 100% RAP. PPB takes advantage of the thermosetting characteristics of PUP resin and chain extender, giving pavement material excellent physical and chemical properties, including high strength, high resilience, fatigue resistance, and so forth [30]. The PPB binder fills the voids in the complex aggregate particles, reducing gradation sensitivity to the asphalt–aggregate ratio and porosity [31]. The demand for new aggregates and new asphalt can be reduced simultaneously. However, such studies can only be found in epoxy asphalt binders up to now. Yi X et al. [32] found that the epoxy asphalt fully recycled mixture was superior to rutting resistance but weak in fatigue performance in contrast to the virgin epoxy asphalt mixture. Hence, the performances, characteristics, and values of the PPB binder are urgent to explore.

This paper is aimed to fabricate a polyurethane-prepolymer-modified bitumen suitable for FRAP and then analyze its feasibility. For this purpose, the content of each component in PPB was determined by different indicators. Then, the effect of thermosetting PUP on the rheological properties and mechanical properties of the asphalt binder was revealed by a dynamic shear rheometer (DSR) and tensile test. Additionally, the effects of the PPB and its FRAP mixture on economic cost were investigated. This paper could be considered a reference study for the research and development of the PPB binder for FRAP.

## 2. Materials and Methods

### 2.1. Raw Materials

#### 2.1.1. Virgin Asphalt

In this study, pen 80/100 virgin asphalt binder was PG 64-22, provided by Inchon, Republic of Korea. Its basic physical parameters were tested according to JTG E20-2011 [33], which is shown in Table 1.

#### 2.1.2. Polyurethane Prepolymer

In this study, the polyurethane prepolymer resin was selected as the asphalt modifier. It was synthesized by polyether polyols and 4,4′-MDI with –NCO content at 10.20~11.20%. Its basic properties are listed in Table 2.

#### 2.1.3. Chain Extender

The 4,4’-bissec-butylamino-diphenylmethane (MDBA) was employed as the chain extender. As a kind of secondary amine chain extender, MDBA can be used to form crosslinking molecular structures by reacting with polyurethane prepolymer, ensuring the mechanical properties of the PPB system. On the other hand, the chemical reaction between polyurethane prepolymer and asphalt occurs moderately with the assistance of secondary amine groups (–NH–) in MDBA. Its physical properties are shown in Table 3.

#### 2.1.4. Compatibilizer

As investigated, there is a great polar difference between polyurethane prepolymer and virgin asphalt binder [34], and it is hard to mix virgin asphalt and modifiers perfectly. In this study, the compatibilizer was added to the PPB system to promote their combination between PUP and virgin asphalt binder. To avoid the influence of compatibilizer on the chemical modification process, maleic anhydride was used as the compatibilizer. Its physical properties are presented in Table 4.

#### 2.1.5. RAP Mixture

RAP was collected from the highway under overhaul where RAP is lacking in coarse aggregates. The gradation of RAP needs adjustment to ensure the performance of FRAP pavement. A 2.36 mm sieve was used to divide RAP into two gradations: RAP smaller than 2.36 mm and RAP greater than 2.36 mm. Through trial and calculations, 20% RAP smaller than 2.36 mm and 80% RAP over 2.36 mm were adopted to form an adjusted RAP, which was close to the gradation range of SMA-13, commonly used in the pavement upper layer. Considering the presence of aged virgin binder in RAP, the asphalt–aggregate ratio of the FRAP mixture was primarily set as 2.0%, and the air void of the FRAP mixture was 4.9%.

### 2.2. Sample Preparation

#### 2.2.1. Preparation Methods of Bitumen

To prepare PPB, an electric mixer and disc sawtooth mixer were selected for this study. The preparation methods of PPB are divided into four steps as follows. Primarily, virgin asphalt and polyurethane prepolymer were put into the oven and preheated for 1 h at 145 °C and 80 °C, respectively, until they were completely melted. Then, the virgin asphalt was stirred by an electric mixer at a stirring speed of 2000 ± 500 rpm/min at 140 °C. Then, the preheated PUP and compatibilizer were added successively in a small amount many times to the unmodified asphalt binder, stirring for 20 ± 5 min, respectively. The chain extender was then added to the mixture and stirred for 5 min, remaining shearing rate and shearing temperature unchanged. Finally, the mixture was kept in an oven for 10~20 min at 140 °C.

#### 2.2.2. Mixing Method of FRAP Mixture

The preparation steps of the FRAP mixture were given as follows. The adjusted RAP was heated in an oven at 140 °C ± 5 °C for 2 h. Then, the virgin binder, PUP, and additives were blended by following the procedures in Section 2.2.1. Next, PPB was added to a mixing machine and mixed with RAP for 90 s. The mixture was compacted using a compaction apparatus after being stirred for 90 s duration. Finally, the test specimens could be prepared after reservation at ambient temperature for 9 days (d).

### 2.3. Test Methods

#### 2.3.1. Brookfield Rotational Viscosity Test

According to the Brookfield viscometer method described in JTG E20-2011 [33], the Brookfield viscosity was measured at 140 °C to evaluate the construction reserved time of PPB. The evolution of Brookfield viscosity with time was tested utilizing rotor 24 # and a shear rate of 20 rpm. All asphalt samples used in the viscosity test were replicated twice.

#### 2.3.2. Tensile Property Test

To investigate the effect of PUP content and chain extender content on the mechanical properties of the PPB binder, the tensile properties of the PPB binder and PUP can be performed by using the Universal Testing Machine. The PPB samples were poured into a silicon rubber mold, cured for 3 d, and a tensile test was followed with the loading rate of PUP and PPB samples at 50 and 500 mm/min, respectively. The tensile samples were produced with 33 mm length, 6 mm width, and 2 mm thickness of the narrow section [35]. Referring to standard ASTM D638-10 [36], their average performances, including tensile strength and elongation at break, were calculated and analyzed to evaluate the strength and toughness of the PPB binder and PUP [37,38]. Considering the asphalt was served in normal climate and pressure and rules stipulated by ASTM D638-2014 [36], the tensile test was conducted at 23 ± 2 °C. Additionally, each sample was replicated three times.

#### 2.3.3. The Marshall Test

In this section, the Marshall test is intended as a confirmatory test to verify the mechanical properties of asphalt mixtures containing 100% RAP. The standard samples Φ101.6 mm × 63.5 mm were made according to JTG E20-2011 [33], and the Marshall test was conducted under a universal testing machine with a loading rate of 50 mm/min. The Marshall stability and flow rate were measured to evaluate the construction quality of the FRAP mixture. Three parallel samples were tested simultaneously.

#### 2.3.4. Segregation Test

The dispersion state of PPB after adding compatibilizer was evaluated by segregation test according to JTG E20-2011 [33]. In this test, about 50 g specimen was poured into an aluminum tube (about 25 mm in diameter, about 140 mm in length, with an opening at one end). The tube was then sealed and placed in an oven for 48 ± 1 h with a storage temperature of 140 °C. After that, the tube was transferred to a refrigerator, cooling at −5 °C for 4 h. Finally, the tube was divided into three pieces, and the softening point of asphalt in the top and bottom of the aluminum tube was tested. The compatibility between modifiers and asphalt was assessed by whether the softening point difference ΔS < 2.5 °C between the top and bottom sections. For each top or bottom section, softening point tests were replicated twice.

#### 2.3.5. Dynamic Shear Rheometer Test

In this research, the rheological behaviors of the PPB binder and virgin binder were characterized using a dynamic shear rheometer. A frequency sweep was performed firstly over a range from 0.1 to 30 Hz with test temperatures ranging from 0 °C to 84 °C. The complex shear modulus *(G*^*^) and phase angle (*δ*) were measured through the frequency sweep test and were utilized to plot master curves based on the time–temperature equivalence principle and the WLF (Williams–Landel–Ferry) equation. Meanwhile, a temperature sweep test was then performed to assess the high-temperature rheological properties according to AASHTO T315-12 [39]. The rutting parameters (*G*^*^/sin *δ*) were calculated with testing temperatures ranging from 46 °C to 82 °C, a temperature interval of 6 °C, and the test frequency fixed at 10 rad/s. Moreover, all asphalt binders used in the DSR test were replicated twice.

## 3. Results

### 3.1. Design of Raw Materials Parameters

#### 3.1.1. PUP and Chain Extender

##### Primary Selection of the Chain Extender Content

Adding a chain extender is aimed at forming crosslinking network structure with PUP, thus ensuring the curing degree and mechanical properties of PUP. The dosage of the chain extender is mainly determined by the molar ratio of key groups in the two components, and Formulas (1) and (2) reveal the chemical reaction process. According to Formula (2), the mass content of MDBA was primarily calculated to be 32.9~36.3% of PUP.

Considering high chain extender content leads to accelerated chemical reaction rates which are not conducive to the construction progress, six different chain extender contents, 22%, 24%, 26%, 28%, 30%, and 32% of PUP weight (short for PUP-22, PUP-24, PUP-26, etc.), were investigated. The tensile stress–strain curves are presented in Figure 1, and the tensile test results of PUP samples are listed in Table 5.
(1)$$OCN-R_{1}-NCO+R_{2}-NH-R_{3}-NH-R_{2}\to R_{1}-N(R_{2})CONH-R_{3}-NHCO(R_{2})N-R_{1}$$
(2)$$W_{c}=W_{\mathrm{PUP}}\times \frac{M_{c}\times n_{c}}{84}\times \mathrm{NCO}\%\times \alpha \times 100\%$$
where *W_c_* is the theoretical chain extender content; *W_PUP_* is PUP content; *M_c_* is the relative molecular weight of the chain extender; *n_c_* is the molar ratio of –NCO/–NH–; NCO% is the isocyanate group content of the prepolymer; α is the chain expansion coefficient; and 84 is the relative molecular weight of 2 isocyanate sets.
(3)$$\mathrm{OCNR}-\mathrm{NCO}+H_{2}O\to HOOCNH-R-\mathrm{NHCOOH}\to NH_{2}-R_{1}-{\mathrm{NH}}_{2}{+\mathrm{CO}}_{2}↑$$

As seen in Figure 1 and Table 5, PUP samples with different chain extender content exhibit a noticeable yield phase. The stress–strain curves generally demonstrate a continuous increase in stress with increasing strain, and plastic deformation can be observed after a brief resilience phase in all samples, indicating their ductile nature [40,41,42].

With the increase of MDBA content, the tensile strength of PUP increases significantly while the elongation at break decreases steadily. One noteworthy observation is that the PUP-32 sample shows the highest tensile strength (10.81 MPa) but the lowest elongation at break (218.32%), while PUP-22 presents the opposite result. This phenomenon is mainly attributed to the increase in crosslinking degree with the extension of MDBA content. The increase in crosslinking degree is caused by the reaction between PUP and MDBA (Formula (1)) and the possible reaction between PUP and moisture (Formula (3)) [43]. Meanwhile, the tensile modulus of PUP grows sharply as the chain extender content rises, demonstrating the increase in rigidity [44]. This may indicate that high chain extender content contributes to the increase in the hard segment. Furthermore, groups from PUP-30 to PUP-32 exhibit a relatively small change in tensile strength and elongation at break, which may result in performance redundancy. Based on the mechanical properties and toughness of the PUP system, it is preliminarily determined that the dosage of chain extender MDBA is controlled within 22~28% and added into asphalt to further determine the proportion of PUP.

##### Principles and Methods for PUP and Chain Extender Content Determination

The content of PUP and chain extender play a key role in the improvement of the properties of PPB binder, ensuring the pavement performance and quality of FRAP. A wide range of factors, including construction reserved time, strength and roughness of asphalt binder, etc., deserve consideration. The larger the PUP and chain extender content, the better the mechanical properties of the PPB binder. However, higher PUP and chain extender content will further accelerate the chemical crosslinking reaction, adversely affecting its working performance and the economic cost of materials production and transportation. Considering the uncertainty of the design parameters above comprehensively, the principles optimization selection method was adopted in this experiment instead of the orthogonal test to reduce the workload of proportional screening. The selection details are as follows.

Firstly, construction reserved time is seen as the primary factor determining the construction feasibility of PPB, which is measured by the Brookfield viscosity test. Referring to GB/T 30598-2014 [45], the Brookfield viscosity of PPB is required to reach 5000 mPa·s not less than 40 min. The time-dependent viscosity effect ensures the working performance of PPB during the construction process. Secondly, the mechanical properties and roughness of asphalt binder are vital factors determining the mechanical properties of the FRAP mixture. According to GB/T 30598-2014 [45], the tensile strength and elongation at break are required to be not less than 1.5 MPa and 200%, respectively, through tensile tests. Nevertheless, excellent mechanical properties require higher tensile strength and lower elongation at break under the premise of relatively lower cost. Finally, the Marshall test is taken into account to verify the construction quality of PPB/FRAP. The Marshall stability of FRAP with PPB binder shall not be less than 20 kN in this case.

Based on the bulk principles above and multiple experimental attempts, four proportions of PUP content are initially determined as 50%, 55%, 60%, and 65% of virgin asphalt weight in this study, which is defined as I, II, III, and IV, respectively. Meanwhile, in accordance with relevant research in 3.1.1.1, chain extender content was determined as 22%, 24%, 26%, and 28% of PUP weight, labeled as A, B, C, and D, respectively (e.g., the PPB sample I-A means that the proportion among virgin asphalt, PUP, and chain extender is equal to 100:50:11). Sixteen ratio groups were determined in total via the single factor variable method. Then, these groups will be selected and identified based on the principles above until obtaining the optimal proportion. Referring to the relevant literature on polyurethane-modified asphalt, the amount of compatibilizer is temporarily fixed at 3% of virgin asphalt weight [46].

##### PUP and Chain Extender Content Determination

For asphalt binders, the viscosity growth rate has a significant effect on asphalt pavement construction. In this study, the Brookfield viscosity test at 140 °C was conducted on the prepared PPB of 16 groups samples, and viscosity curve changes with time were observed in Figure 2.

From Figure 2, it can be seen that the viscosity of PPB rises slowly with time at the initial stage, and then the growth rate rises dramatically, especially 30 min later. This phenomenon may attribute to autocatalytic characteristics, which means that the reaction is catalyzed by the reaction product without adding a catalyst.

Comparing the viscosity results of the 16 groups, it can be observed that most PPB binders in groups I, II, and III reach 5000 mPa·s or higher for more than 40 min. Group I-A exhibits the longest retention time, lasting 81 min, whereas the figure for group II-D is the shortest (42 min). These findings indicate that higher PUP and chain extender content can accelerate the progression of the chemical reaction. Moreover, there is a more evident difference among group I, II, III, and IV rather than groups A, B, C, and D, indicating that PUP has a greater impact on chemical reaction rate than that of the chain extender. Furthermore, groups III-D, IV-C, and IV-D are excluded due to their excessive reaction rates.

The remaining 13 groups are left for mechanical property test of PPB, and relevant analysis is expected to further select the optimal group. The effect of PUP chain extender content on the mechanical properties is studied via a tensile test, the result of which is shown in Figure 3. As investigated in Figure 2, since the performance of the PPB binder can be more easily affected by PUP content in contrast to chain extender content, the proportion of the chain extender is fixed to study the variation of the tensile properties.

As investigated, the tensile strength goes up with the increase of PUP content, while there is an opposite trend for the elongation at break. Take group A (as seen in Figure 3a) as an example: as the PUP content increases from 50% to 65%, the tensile strength of the entire system increases greatly while the elongation at break decreases gently. However, the PPB binders in group A are weak in tensile properties even if their elongation at break is brilliant. Moreover, group III-C exhibits the optimal mechanical property with its tensile strength and elongation at break, reaching 4.52 MPa and 308.34%, respectively. It is particularly noticeable that in terms of group D, the tensile strength increases by 19.2%, and the elongation at break decreases sharply by 28.2% when the PUP content increases from 50% to 55%. Generally, the skeleton formed by the reaction between PUP and MDBA contributes to the improvement in the tensile strength of PPB. However, the addition of PUP results in a reduction in asphalt content, which yet impairs the flexibility of the PPB binder.

Combined with the basic requirements of the tensile test in GB/T 30598-2014 [45] (tensile strength ≥ 1.5 MPa, elongation at break ≥ 200%), group IV-B is selected as the superior proportion in group B. Similarly, groups III-C and II-D were chosen as the optimized group as well.

To verify the mechanical indicators of the fully recycled mixture, the Marshall test is adopted in this section based on JTG E20-2011 T0709 [33]; the test results are shown in Figure 4. It can be found that the Marshall stabilities of the three PPB/FRAP samples exceed 20 kN. Among them, the stability of group III-C is the highest, reaching 32.39 kN, whereas the figure for group II-D is the minimum (24.04 kN), which presents superior mechanical properties. However, for groups II-D and IV-B, their flow values are slightly out of the normal range (15~45 dmm), indicating that they are more susceptible to vertical deformations when damaged by compression. Marshall test results reveal that the PUP component could greatly reduce the impact of aggregate and aged asphalt on the FRAP mixture. In general, the PPB binder provides a feasible development direction for fully reclaimed asphalt pavement, and III-C is selected as the optimal proportion in the PPB binder system.

#### 3.1.2. Compatibilizer

The purpose of incorporating the compatibilizer is to improve the dispersion state and storage stability of PUP in the PPB system and then reduce the segregation between modifiers and asphalt during storage and usage. To diminish the polar difference between PUP resin and the virgin binder, the segregation test was employed to determine the compatibility quality in this study. The compatibilizer with different content, 1%, 2%, 3%, 4%, and 5%, was chosen to explore their softening point differences; the results are shown in Figure 5.

As can be seen in Figure 5, the softening point difference between the top and bottom sections (Δ*S*) dramatically decreases with the addition of the compatibilizer. When there was no compatibilizer added, significant segregation occurred between the PUP and virgin binder, with ΔS reaching 5.3 °C. The Δ*S* value decreases constantly with the compatibilizer content increasing, and only the 3%, 4%, and 5% compatibilizer content meet the requirement of Δ*S* < 2.5 °C. Combined with the construction cost of the raw materials, 3% was determined as the appropriate compatibilizer content, and the prepared asphalt has good compatibility with PUP.

### 3.2. Study on Rheological Properties of PPB

In this study, virgin asphalt binder and PPB asphalt binder with different proportions (50%, 55%, 60%, and 65%) were selected as the research objects, corresponding to virgin asphalt, I-C, II-C, III-C, IV-C in Section 3.1, respectively.

#### 3.2.1. Frequency Sweep Test Results

In this test, the master curves of PPB samples with different PUP content in the linear viscoelastic range were obtained and analyzed by a frequency sweep test. In accordance with the CAM model, the complex shear modulus and phase angle master curves were constructed based on the time–temperature equivalence principle and the WLF (Williams–Landel–Ferry) equation, illustrated in Figure 6. The reference temperature was set to 36 °C. CAM model formula was shown as follows.
(4)$$\left|G^{*}(f,T)\right|=\frac{G_{g}}{{\left[1+{\left(f_{c}/\alpha_{T}f\right)}^{k}\right]}^{m_{e}/k}}$$
where *f* is the reduced frequency, *T* is the reference temperature at which the master curve is developed, *G_g_* is the glass shear modulus, *f*_c_ is the position parameter, *k* and *m_e_* are the shape parameters of dimension 1, and *α_T_* is a shift factor related to WLF equations.

The amount of energy for the asphalt binder to withstand deformation is measured by the complex shear modulus (*G*^*^). From its master curves, it can be found that the incorporation of PUP significantly increases the *G*^*^ value compared to the virgin binder, and the increased magnitude decreases as PUP content increases. This result indicates that the PUP exhibits superior elastic properties and contributes to the increase of the elastic component in the PPB system. Consequently, the addition of PUP enhances the deformation resistance under heavy loads and lowers the temperature sensitivity. Moreover, the increased range of the complex modulus of PPB is significantly higher than that of the virgin binder in low-frequency region (10^−4^~10^2^ Hz). Specifically, the increase in *G*^*^ at 65% PUP is particularly significant compared to the sample with 50% PUP content. This result implies that the hardness and the rutting resistance of PPB are greatly improved.

The phase angle is an indicator reflecting the proportion of elastic and viscous components in bitumen. As seen in Figure 6b, the phase angle of the PPB binder shows a downward trend with the extension of PUP content. This trend is particularly evident in the low-frequency zone, further demonstrating that the use of PUP improves the elastic recovery property of virgin asphalt. Meanwhile, the δ master curves exhibit significant fluctuations and plateaus ranging from 10^−3^ to 10^0^ Hz. This behavior suggests the occurrence of a phase transition in this frequency zone, providing evidence for the formation of a crosslinking grid structure between PUP and asphalt [47,48]. It is worth noting that a higher *G*^*^ and a lower *δ* imply less energy loss under repeated loads. Hence, the addition of PUP affects the fatigue performance of PPB to some extent. In summary, the PPB exhibits a higher *G*^*^ and lower *δ* over the entire frequency range, indicating the reduction in the temperature susceptibility as well as fatigue performance of the modified asphalt.

#### 3.2.2. Temperature Sweep TEST results

The rutting factor (*G*^*^/sin *δ*) represents the rutting resistance of asphalt under high-temperature conditions. In this case, the higher value of *G*^*^/sin *δ* implies greater rutting resistance.

The temperature sweep test results of the PPB are summarized in Figure 7. It can be found that a rapid fall occurs in the *G*^*^/sin *δ* curve of the PPB samples with temperature increases, and the differences between different PUP groups are obvious. As shown, the rutting factors of all PPB samples are higher than that of the virgin binder, implying an improvement in the high-temperature performance of asphalt with the incorporation of PUP. In detail, the rutting factor shows an upward trend as PUP content increases. For instance, the rutting factor of group IV-C is twice as large as that of group I-C at 64 °C. This suggests that the modification effect on high-temperature performance becomes more pronounced with higher PUP content. Notably, the rutting factor experiences a significant increase when the PUP content increases from 55% to 65%, exhibiting excellent high-temperature deformation resistance for higher PUP content. In summary, the PUP plays a critical role in improving the high-temperature rutting resistance of the PPB binder. Combined with the analysis results in Figure 6, it is easy to find that the significant increase in the rutting factor is largely dependent on the increase in complex modulus.

### 3.3. Study on Tensile Properties of PPB

The curing degree of the PUP system, as a thermosetting resin, plays a crucial role in determining the mechanical properties of the PPB binder. To ensure the pavement performance of the PPB binder used in the FRAP mixture, the effect of curing time on the mechanical property was investigated through the tensile test for up to 15 days of curing at ambient temperature. The variation of the tensile property of the PPB binder as a function of curing time was illustrated in Figure 8.

As shown, the tensile strength of the PPB binder undergoes a dramatic increase with curing time in the initial 3 days, increasing from 0.35 MPa to 2.45 MPa due to the rapid crosslinking of the PPB system. As the curing process proceeds, the growth rate of tensile strength slows down in the following 6 days and eventually stabilizes after 9-day curing. The steady value was calculated to be 4.09 ± 0.22 MPa, which is 1.67 times and 1.19 times higher than those of samples cured for 3 d and 7 d, respectively. During the crosslinking process, the isocyanate group primarily reacts with the chain extender, followed by a chemical reaction with the light component in the asphalt. Among these urethane bonds transformed by the isocyanate group, hydrogen bonds were formed, and physical crosslinking points were gradually created in the asphalt system [49]. These reactions contribute to the compaction of the crosslinking structure in the PPB system, resulting in the initial increment in tensile properties. Then, the curing speed decreases mainly because the polymerization between PUP and the chain extender tends to be complete, and only a small amount of excessive PUP reacts with the moisture in the air. After 9 d curing, the remaining reaction becomes weaker as the network structure is formed. Therefore, it is necessary for the PPB binder used in the FRAP mixture to be conserved for not less than 9 d to ensure its mechanical property. The curing process can be carried out in a natural environment, and its properties will be gradually improved along with the curing.

### 3.4. Economic Cost Impact Evaluation

Apart from excellent performance and convenient preparation methods, the costs of PPB binder and FRAP are one of the key factors determining the application of fully recycling technology. A production cost comparison was conducted between the PPB binder, epoxy asphalt, SBS-modified asphalt, and crumb-rubber-modified asphalt based on the formula of modified binders and the unit price of each component, as presented in Figure 9a. All the data of unit prices were investigated based on the market survey. The prices of raw materials and mass ratios of various asphalt binders and mixtures are listed in Table 6. As shown, it is obvious that the unit price of the PPB binder is relatively lower than that of epoxy asphalt but far exceeds that of the SBS-modified asphalt (MA) binder. The cost of the PPB binder (ca. 1963.26 $/t) is 77.30% of that of the epoxy-modified binder (ca. 2539.66 $/t), exhibiting a dramatic price advantage as compared to the epoxy-modified binder. Meanwhile, the SBS MA and crumb rubber MA binder present a low cost, mainly due to a relatively small content. Particularly, the production cost of crumb rubber MA is even lower than that of virgin binder because of the low cost of crumb rubber. 

The cost of different mixtures for paving applications was calculated and compared, including the PPB/FRAP mixture, PPB/AC-13, and PPB/SMA-13, as shown in Figure 9b. It was demonstrated that the unit price of the PPB/FRAP mixture was greatly lower than the figure for the PPB mixture without RAP. When PPB binder is applied to fully reclaimed asphalt pavement, the aged asphalt contained in RAP softens and blends with the PPB. Therefore, the PPB content in the FRAP mixture is lowered indirectly, thus lowering cost further. As such, a low asphalt–aggregate ratio and 100% RAP further contribute to the reduction of construction costs.

## 4. Conclusions

A polyurethane-prepolymer-modified bitumen (PPB) binder is studied primarily from preparation formula, preparation method, performances, and economic value. It has been proved that the PPB binder is feasible to be the binder for the fully reclaimed asphalt pavement (FRAP) mixture. Based on the results and discussion above, the following conclusions can be drawn:(a)The PPB was obtained by chemical synthesis containing simply raw materials whose preparation process has the advantages of relatively low shearing temperature, short time, and good compatibility between virgin asphalt and modifiers. The appropriate formula of PPB binder is determined, i.e., virgin asphalt binder: PUP: chain extender: compatibilizer = 100:60:15.6:3;(b)In terms of the rheological properties, the addition of PUP significantly improved the elastic recovery performance, high-temperature permanent deformation resistance, and temperature sensitivity of asphalt but weakened the influence on fatigue resistance;(c)The great tensile properties of the PPB binder result from the crosslinking network structure formed by PUP resin curing. The network structure makes asphalt has better pavement performance which is conducive to forming strength;(d)The PPB mixtures containing 100% RAP show potential cost-saving benefits compared to epoxy asphalt mixtures. The unit price of PPB binder is 77.30% of that of epoxy asphalt binder, and the utilization of RAP further reduces the cost of raw materials. This demonstrates its suitability for fully reclaimed asphalt pavement applications, where the application of recycled materials is maximized.

Based on the conclusions above, it can be inferred that the PPB binder is a viable solution for enhancing the performance and economic value of asphalt pavement recycling technologies. Its synthesis process, improved properties, and cost-saving potential make it a promising choice for fully reclaimed asphalt pavement projects. To further reveal the feasibility of fully reclaimed asphalt pavement applications, the next work will focus on the performance enhancement mechanism of PPB, the PPB-RAP interface interaction, fatigue properties, and the durability of the FRAP mixture.

## Figures and Tables

**Figure 1 materials-16-05686-f001:**
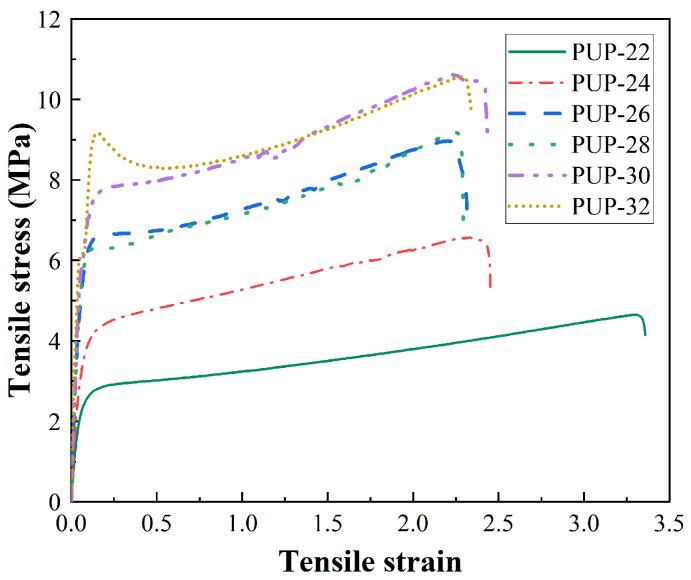
The tensile stress–strain curves of PUP samples.

**Figure 2 materials-16-05686-f002:**
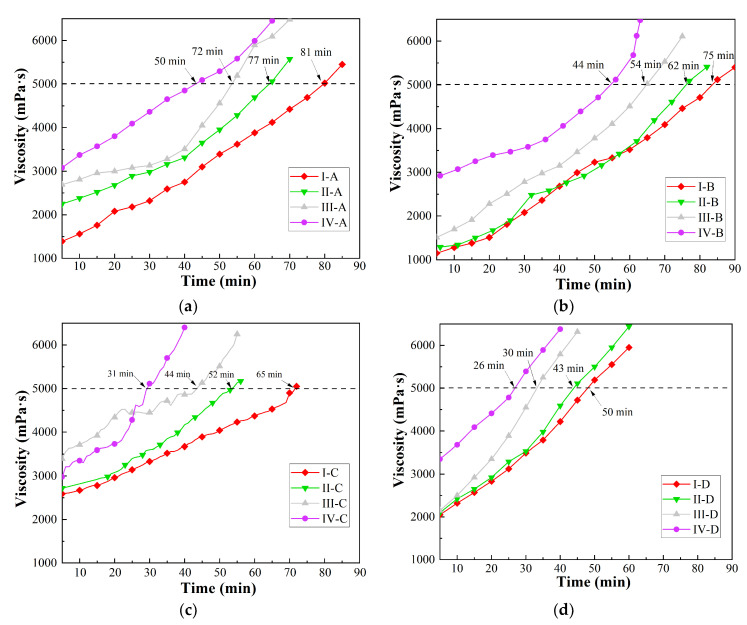
Curves of the Brookfield rotational viscosity–time with different PUP content: (**a**) group A; (**b**) group B; (**c**) group C; (**d**) group D.

**Figure 3 materials-16-05686-f003:**
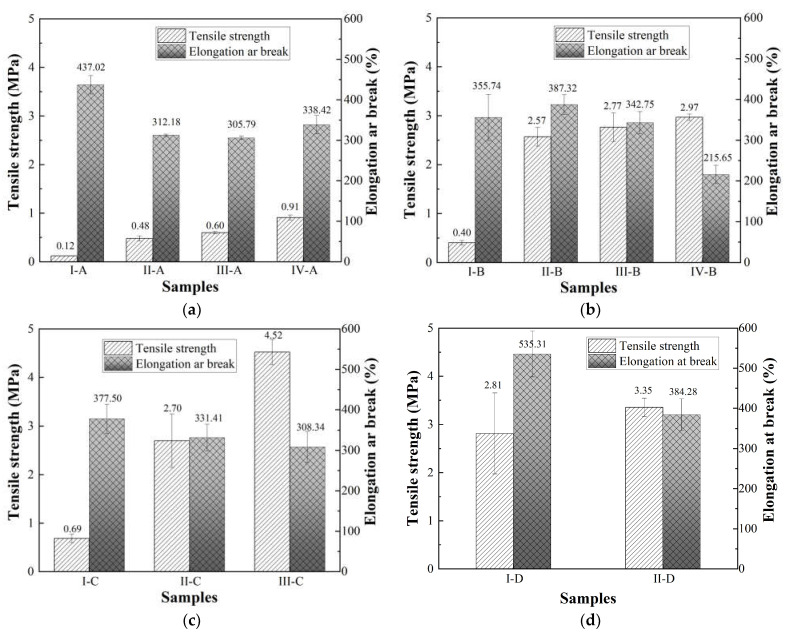
The tensile test results of the PPB binder with different PUP content: (**a**) group A; (**b**) group B; (**c**) group C; (**d**) group D.

**Figure 4 materials-16-05686-f004:**
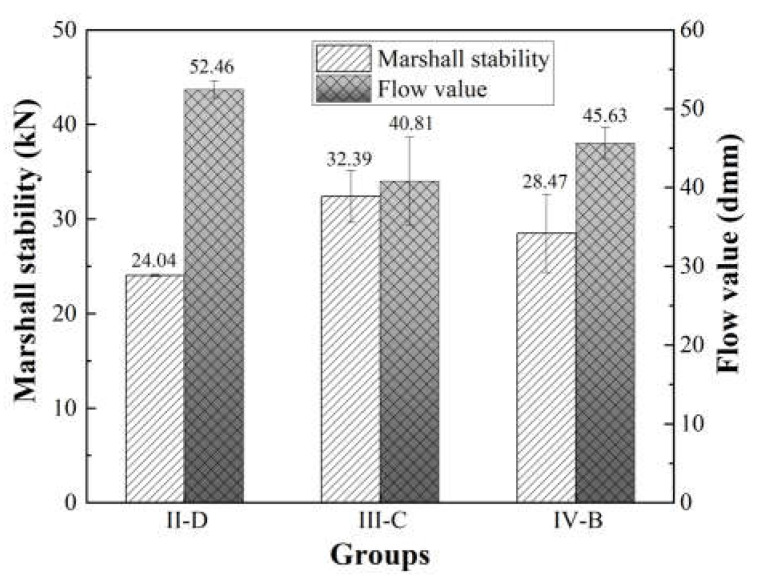
Marshall test results.

**Figure 5 materials-16-05686-f005:**
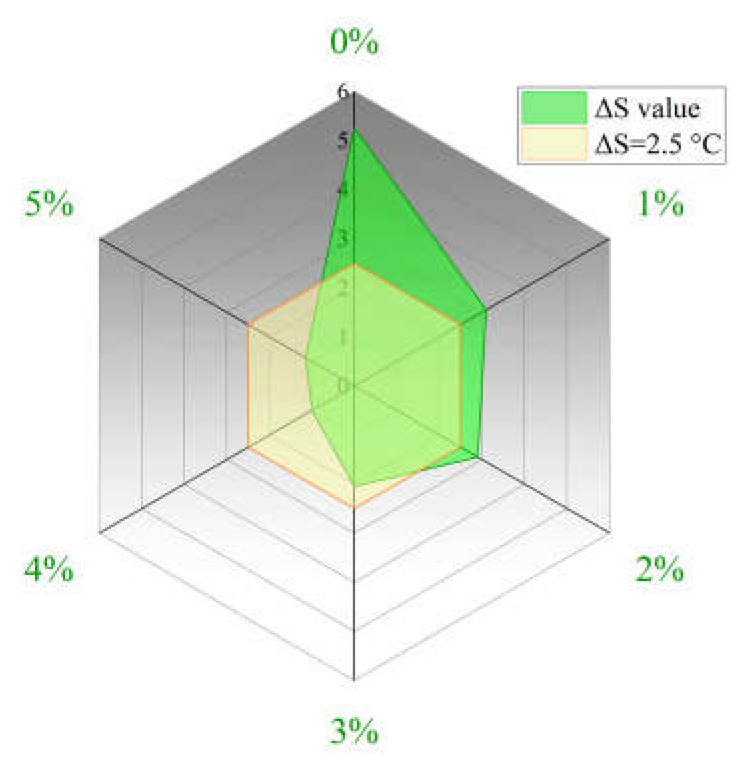
ΔS of PPB binders with different compatibilizer content.

**Figure 6 materials-16-05686-f006:**
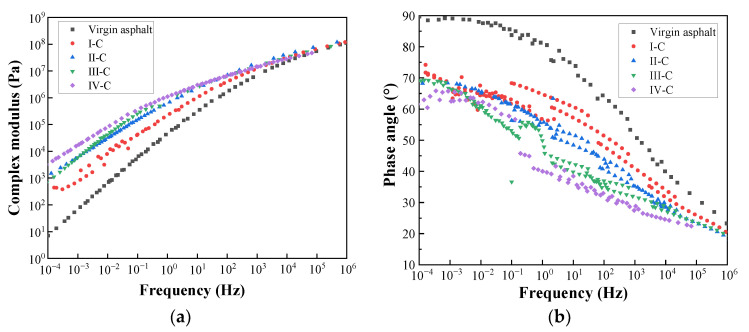
The master curves of PPB binder with different PUP content: (**a**) complex shear modulus; (**b**) phase angle.

**Figure 7 materials-16-05686-f007:**
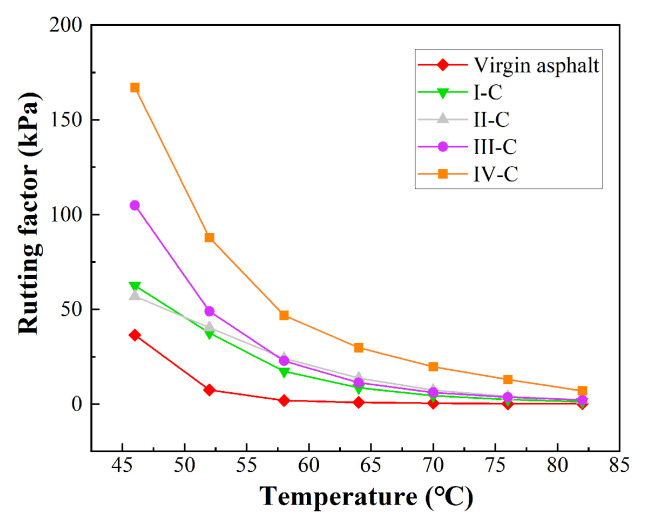
The temperature sweep test results.

**Figure 8 materials-16-05686-f008:**
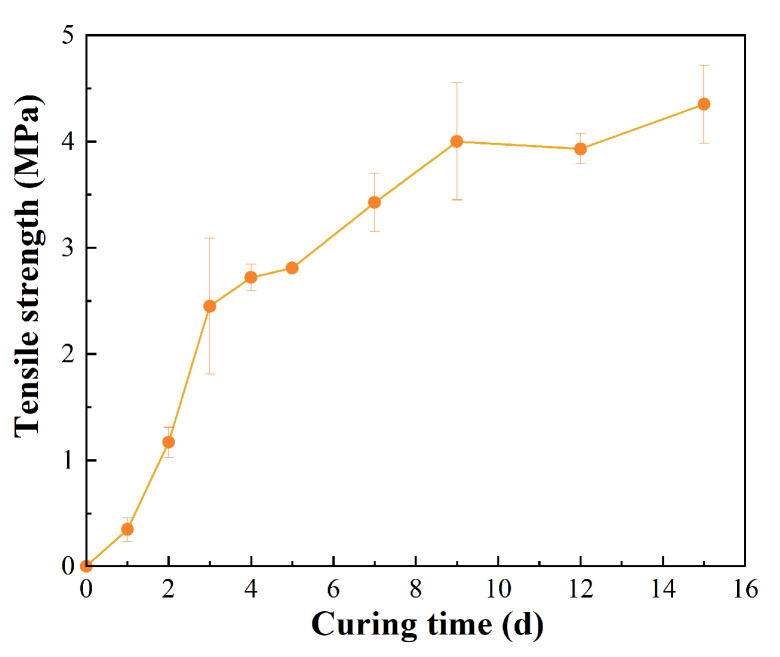
Evolutions in the tensile property of the PPB binder.

**Figure 9 materials-16-05686-f009:**
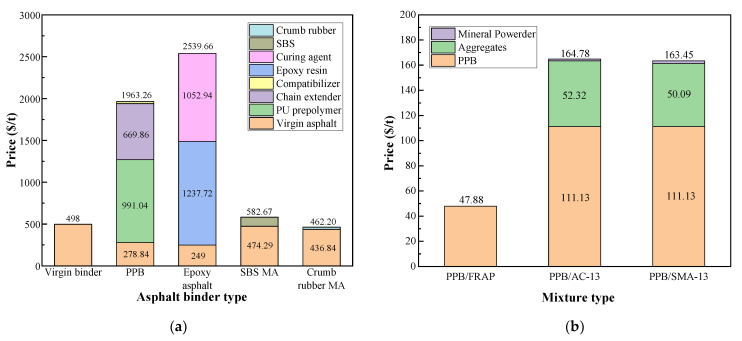
(**a**) Cost comparison of different modified asphalt binders; (**b**) cost comparison of different asphalt mixtures [50,51,52] ($1 = ￥6.78, 2023).

**Table 1 materials-16-05686-t001:** Basic information of SK-90# virgin asphalt.

Properties	Units	Measured Value	Requirement	Specification
Penetration	0.1 mm	80.3	80~100	T 0604-2011
Softening point	°C	48.4	≥46	T 0606-2011
Ductility (15 °C)	cm	143	≥100	T 0605-2011
Viscosity (135 °C)	mPa·s	413	Measured records	T 0620-2011
Mass loss after TFOT	%	−0.69	≤±0.8	T 0610-2011

**Table 2 materials-16-05686-t002:** Basic information of polyurethane prepolymer.

Properties	Units	Results
Appearance	/	Transparent
Viscosity (25 °C)	mPa·s	2200~3200
Density	g/cm^3^	1.05~1.10
–NCO content	%	10.20~11.20

**Table 3 materials-16-05686-t003:** Basic properties of chain extender.

Properties	Units	Results
Appearance	/	Dark amber liquid
Molecular formula	/	C_30_H_30_N_2_
Molecular weight	/	310.48
Density	g/cm^3^	0.99
Purity	%	98.70
MDA content	%	0.01
Moisture content	%	0.018

**Table 4 materials-16-05686-t004:** Basic properties of compatibilizer.

Properties	Units	Results
Appearance	/	White powder crystal
Melting point	°C	54
Boiling point	°C	202
Density	g/cm^3^	1.48

**Table 5 materials-16-05686-t005:** Tensile properties of PUP samples.

No.	Tensile Strength/MPa	Elongation at Break/%	Tensile Modulus/MPa
PUP-22	5.14 ± 0.46	315.59 ± 32.67	16.70 ± 3.00
PUP-24	8.47 ± 0.68	255.03 ± 6.95	28.61 ± 7.03
PUP-26	8.72 ± 0.37	252.82 ± 39.16	33.37 ± 3.85
PUP-28	9.73 ± 0.77	249.96 ± 29.45	34.78 ± 0.67
PUP-30	10.42 ± 0.27	243.28 ± 40.33	40.46 ± 3.45
PUP-32	10.81 ± 0.50	218.32 ± 26.19	42.60 ± 8.06

**Table 6 materials-16-05686-t006:** The prices of raw materials and mass ratios of various asphalt binders and mixtures.

Materials Types	Raw Materials	Formulation
Asphalt binder type	Virgin asphalt 498 $/t; PUP 2950 $/t; Chain extender 7669 $/t; Compatibilizer 1401 $/t; Epoxy resin 4268 $/t; Curing agent 5014 $/t; SBS 108.38 $/t; Crumb rubber 25.36 $/t.	PPB: virgin asphalt: PUP: chain extender: compatibilizer = 100:60:15.6:3; Epoxy asphalt: virgin asphalt: epoxy resin: curing agent = 100:58:42; SBS MA: virgin asphalt: SBS = 100:5; Crumb rubber MA: virgin asphalt: crumb rubber = 100:14.
Mixture type	PPB binder 1963.26 $/t; Aggregates 59 $/t; Mineral powder 23.6 $/t; RAP 0 $/t.	PPB/FRAP: 2.44%PPB, 97.56%RAP; PPB/AC-13: 5.66%PPB, 88.68%aggregates, 5.66%mineral powder; PPB/SMA-13: 5.66%PPB, 84.91%aggregates, 9.43%mineral powder.

## Data Availability

The data that support the findings of this study are available upon reasonable request from the authors.

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
