# Peer review of "Feasibility Analysis of Polyurethane-Prepolymer-Modified Bitumen Used for Fully Reclaimed Asphalt Pavement (FRAP)"

_materials, 2023, doi:10.3390/ma16165686_

Round 1
Reviewer 1 Report
The study is well-designed, the employed methods are discussed well and the data are supported by the results provided. There are some minor drawbacks that have to be modified which they listed in the file attached.

Extensive editing of the English language required
Reviewer 2 Report
The results and discussion are complete and I enjoyed reading the manuscript. But I suggest you study the tensile properties at some other temperatures. As reported by the authors, the test temperature is 23 °C. As we know, temperature is very important for polymers because the properties and behavior of polymers change with the change in temperature. Certainly, the behavior of samples at higher or lower temperatures is different, which is important. There are always climate changes and if the sample that is prepared is intended for industrial use, it is better to be more careful about it, although this does not reduce the value of your study. I suggest you add this section (Temperature change for tensile testing) to this study, although this can cause extensive changes in this manuscript, so if you can't, add a paragraph on this topic and explain the reason for choosing the temperature of 23 °C. also, the language of the manuscript should be checked.
Moderate editing of English language required.
Reviewer 3 Report
The manuscript concerns the use of polyurethane prepolymer modified bitumen in fully reclaimed asphalt pavement. It is an important topic, and the authors have done good research. The manuscript describes an experimental study developed at the laboratory with an asphalt mixture SMA13.
Some suggestions could help improve the paper.
Please, confirm that all the standard methods presented in the manuscript are included as references.
It needs to be clarified what the precision is for all the experimental results (e.g., repeatability). The number of specimens of each sample used to evaluate the average and the standard deviation of results should be include in the manuscript.
The information about the composition and properties of the asphalt mixture needs to be more comprehensive for a better understanding of the behaviour of the material (e.g., Marshall properties).
Do the authors think that it is good performance for recycled mixtures to have high Marshall stabilities? What is the opinion of the authors concerning fatigue and stiffness behaviours? What will be the durability of the asphalt mixture? If the main objective is to present the influence of the polyurethane prepolymer modified bitumen in the asphalt mixture, the authors should include more information about the asphalt mixture behaviour. Another alternative is to focus the paper on the PPB binder behaviour, excluding the information concerning the RAP mixture that could be presented in another submission.
It could be interesting to present some discussion on future works in Section 4 (conclusions).
Reviewer 4 Report
There are significant inconsistencies between the response to the reviewers and the revised manuscript. The Authors should carefully check that all the declared changes are actually implemented in the paper.
Some examples are the following:
- Ref 16 is wrong in the reference list. Please, add the right reference in the reference list and check the correspondence between all references cited in the text and the bibliography.
- The preparation of the asphalt concrete mixtures (mixing, compaction, coring, air voids, etc.) is still missing in the paper.
- Eq. 3: aT is still named "displacement factor" in the paper.
- The Authors comment the phase angle reduction in a biased way. The concurrent increase of stiffness and decrease of phase angle can imply a worsening of the fatigue behaviour. This aspect was not investigated by the Authors, thus this possible negative consequence on the material fatigue behaviour should be discussed in the paper.
- The master curve figure in the paper is different from that in the response to the reviewers.
- Final considerations have not been added at the end of the conclusion section (contrarily to what declared by the Authors in their response).
Extensive editing of English language required
Round 2
Reviewer 3 Report
The authors have addressed the main recommendations. Congratulations on the research and the manuscript.
Author Response
Thanks for your comments.
